# A Brief Review of Meiotic Chromosomes in Early Spermatogenesis and Oogenesis and Mitotic Chromosomes in the Viviparous Lizard *Zootoca vivipara* (Squamata: Lacertidae) with Multiple Sex Chromosomes

**DOI:** 10.3390/ani13010019

**Published:** 2022-12-20

**Authors:** Larissa Kurpyianova, Larissa Safronova

**Affiliations:** 1Zoological Institute of the Russian Academy of Sciences (ZIN), 199034 Saint Petersburg, Russia; 2Severtsov Institute of Ecology and Evolution, Russian Academy of Sciences, 119071 Moscow, Russia

**Keywords:** lizards, *Zootoca vivipara*, multiple sex chromosomes, meiosis, synaptonemal complex (SC), form and subspeciation

## Abstract

**Simple Summary:**

The wide-ranging Eurasian species *Zootoca vivipara* (Lichtenstein, 1823), of the family Lacertidae (Reptilia), is a rare species within the family, possessing multiple sex chromosomes (male Z_1_Z_2_Z_1_Z_2_/Z_1_Z_2_W female). In addition, the intense reorganization of this W sex chromosome is accompanied by active subspeciation and the formation of 4–5 cryptic taxa. In the females of two cryptic forms having a similar system of multiple sex chromosomes (Z_1_Z_2_W) but with different morphology, the cytogenetic and specific genomic structures of the W sex chromosome’s early oogenesis and meiosis have standard occurrence. Despite the ambiguous behavior of the three presumed sex chromosomes at the early stages of meiotic prophase I, variability in their number of bivalents and chromosomes and significant disturbances in chromosome segregation have not been discovered. Because in *Z. vivipara* the W sex chromosome, unlike all the other chromosomes, does not have several identified SINE-Zv and TE elements, we may assume that the specific genomic structure of this chromosome may be one of the factors ensuring meiotic stability in the cryptic taxa of the species with the multiple sex chromosomes. The question of female meiotic drive in the meiosis of the cryptic forms of the *Z. vivipara* complex is still obscure.

**Abstract:**

This brief review is focused on the viviparous lizard *Zootoca vivipara* (Lichtenstein, 1823), of the family Lacertidae, which possesses female heterogamety and multiple sex chromosomes (male 2*n* = 36, Z_1_Z_1_Z_2_Z_2_/Z_1_Z_2_W, female 2*n* = 35, with variable W sex chromosome). Multiple sex chromosomes and their changes may influence meiosis and the female meiotic drive, and they may play a role in reproductive isolation. In two cryptic taxa of *Z. vivipara* with different W sex chromosomes, meiosis during early spermatogenesis and oogenesis proceeds normally, without any disturbances, with the formation of haploid spermatocytes, and in female meiosis with the formation of synaptonemal complexes (SCs) and the lampbrush chromosomes. In females, the SC number was constantly equal to 19 (according to the SC length, 16 SC autosomal bivalents plus three presumed SC sex chromosome elements). No variability in the chromosomes at the early stages of meiotic prophase I, and no significant disturbances in the chromosome segregation at the anaphase–telophase I stage, have been discovered, and haploid oocytes (*n* = 17) at the metaphase II stage have been revealed. There should be a factor/factors that maintain the multiple sex chromosomes, their equal transmission, and the course of meiosis in these cryptic forms of *Z. vivipara.*

## 1. Introduction

Lizards are one of a few groups of reptiles whose members are characterized by temperature (TSD) and genetic (GSD) sex determination (e.g., Reference [1]). Male and female heterogamety (XY/ZW) and a variety of sex chromosome systems are also found in these animals [1,2,3]. About 23% of karyotyped lizard species have multiple sex chromosomes [3]. This system of sex chromosomes is usually common in XY but not in ZW groups. Presumably, as suggested by some authors [2], it is due to the different involvement of sex-specific sex chromosomes in female meiosis, and the effect of female meiotic drive. For instance, mammals possessing male heterogamety show many taxa with multiple sex chromosomes [4]. In contrast, birds (ZW), except for one species [5], do not display multiple sex chromosomes [6].

Lizards of the family Lacertidae, except for a few debated cases [7], have genetic sex determination, demonstrate female heterogamety (ZW), and only four of the karyotyped lacertid species (totaling about 115) reveal multiple sex chromosomes Z_1_Z_2_W [3]. During the evolution of one of those species, namely, *Zootoca vivipara* (Lichtenstein, 1823), the acrocentric (A) sex chromosome Z was involved in a rearrangement, i.e., in translocation with an autosomal acrocentric, which led to the creation of multiple sex chromosomes Z_1_Z_2_W and a completed W sex macrochromosome comprising the Z_1_ and Z_2_ chromosomes. The sex/autosome fusion has also been described among other lizard species, for example, in the Chamaeleonidae [8]. The widely distributed Eurasian lizard *Z. vivipara* is characterized by some other special features: both multiple sex chromosomes Z_1_Z_2_W and simple Zw system, and oviparous/viviparous reproduction in different populations. The species is polymorphic on its mitochondrial (mt) haplotype but considerably uniform morphologically; moreover, viviparous *Z. vivipara* shows considerable diversity in the morphology and structure of the W sex chromosome [9,10,11,12,13]. The intense reorganization of the W sex chromosome appears to be accompanied by active subspeciation and speciation, and by the formation of cryptic taxa with different W chromosomes (Table 1) [12,13]. Several chromosomal forms and subspecies of *Z. vivipara* have their distinct distribution areas (allopatric and parapatric populations) in the central and southern-central part of Europe. Some of them have a mosaic pattern of populations, and inhabit small areas, while others are relict forms [13]. All the specimens of *Z. vivipara* can be diagnosed and recognized by their mt haplotype [14,15] and by some chromosomal characteristics, in particular, by the morphology and the cytogenetical structure (the amount and distribution of heterochromatin, C-bands) of their W sex chromosome [11,13,16]. It should be noticed that the two forms, the western form and the eastern (Russian) form of *Z. vivipara*, occupy a vast territory in Europe and Asia. The western form inhabits central and western Europe, whereas the eastern (Russian) form populates eastern Europe and Asia. In addition, both of these forms of *Z. vivipara* have been discovered in the Baltic region (Figure 1, map) [17,18].

Thus, it is clear that karyological differentiation in the *Z. vivipara* complex is high, and in particular in the morphology and the cytogenetical structure (the amount and distribution of heterochromatin, C-bands) of their W sex chromosome. As is well known, chromosome reorganization may play an important role in sex chromosome differentiation in different lizard groups [3]. The role of variable sex chromosomes in the evolution of the cryptic *Z. vivipara* complex is still poorly studied, as is the connection between sex chromosomes and reproductive isolation. According to King [19,20], simple or multiple sex chromosomes may play a role in meiosis (as a reproductive meiotic barrier) and the speciation of different groups. Moreover, reproductive isolation seems to evolve faster among species with heteromorphic sex chromosomes [21,22]. Alterations in some patterns of multiple sex chromosomes, among other things, may reinforce their isolation effect.

Furthermore, genomic composition is a factor favoring the fixation of mutant karyotypes [23]. In addition to the variable multiple sex chromosomes in the karyotype of *Z. vivipara*, several new molecular markers, namely, some different short interspersed elements (SINEs) and transposable elements (TE), have recently been detected and identified in its genome [24]. SINE elements, as is well known, often have preferred sites in the genome and may also influence the process of meiosis, speciation rate, etc.

All the mentioned characteristics offer a rare possibility to use *Z. vivipara* as a model for studying some general evolutionary problems; for instance, sex chromosome evolution and its impact on subspeciation and form formation. Given the above, the karyotype, especially the characteristics of sex chromosomes, meiosis, and behavior of chromosomes during the meiosis of the described cryptic taxa with variable multiple sex chromosomes, are of particular interest.

In this brief review, we consider mainly the features of karyotype and sex chromosomes, and the course of spermato- and oogenesis, meiosis, and the behavior of chromosomes in the early stages of prophase I of meiosis, in two closely related cryptic chromosomal forms of *Z. vivipara* with multiple sex chromosomes Z_1_Z_2_W: the eastern (Russian) form and the western form from the Baltic region of Russia (Figure 1, map).

They have the viviparous mode of reproduction and are diverse in their karyotypes, namely, the morphology and cytogenetical structure of the W sex chromosomes [17,25,26]. Several samples of the *Z. vivipara* under study have already been used in other studies. The species is not included in the national Red Data Book and lists of protected taxa. It is not included in international agreements. The specimens were treated by ether according to ethical practices and were deposited in the collection of ZISP, chromosomal collection, accession numbers №№ 9261–9263; 9448–9450. Chromosomal material and the preparations from oocytes were stored in a freezer (minus 22–25 °C) and some of them were used in this work for the first time. A total of six males and females of *Z. vivipara* were collected in the Leningrad and Kaliningrad areas (Baltic region, Russia). Chromosomal preparations had been obtained by the scraping and air-drying method from intestine, gonads, and germinal lamina cells, and then they were stained with Giemsa. C-banding was carried out according to Summer’s method [27], and fluorochrome AT staining (DAPI) using the method of Drs. M. Schmid and M. Guttembach [28]. Meiotic preparations were obtained by using the method of total oocyte nuclei spreading developed by M. Dresser and M. Moses [29]. Chromosome preparations were stained with Giemsa, and for the visualization of synaptonemal complexes (SCs), total preparations were stained with silver nitrate and DAPI. Fluorescent analysis with the help of incubation with primary and secondary antibodies SYPC3 (the protein of synaptonemal complexes (SC) of central elements), and fluorochrome AT DAPI staining was performed on the preparations. The lengths of the SCs of bivalents were measured using Leica Application Suite V3 on the digital microphotographs. The SCs of the bivalents in a karyotype were numbered in the decreasing order of their linear sizes. Analysis of the photos and SC karyotyping were conducted on the basis of the measurements of the SC by the relative lengths of each individual SC.

## 2. Characteristics of Karyotype

In the karyotype markers, these forms of *Z. vivipara* are characterized by different diploid chromosome numbers (male 2*n* = 36 acrocentrics (A) and female 2*n* = 35) and different numbers of sex chromosomes (in male Z_1_Z_1_Z_2_Z_2_ and Z_1_Z_2_W in female), with different morphology of the W sex chromosome (the eastern (Russian) cryptic form W is acro/subtelocentric (A/ST) and the western cryptic form W is submetacentric (SV)) [11,12,17,18]. Thus, the male karyotype is 2*n* = 36 A: 32A + Z_1_Z_1_Z_2_Z_2_, while the female karyotype is 2*n* = 35: 32 A+ Z_1_Z_2_W, where W is (A/ST) or (SV). Further comparative staining analyses of C-banding/CMA_3_/DAPI have also shown the different cytogenetic structure (the distribution of conspicuous centromeric and telomeric C-bands), and the presence of an additional interstitial C-positive heterochromatin block, by staining with an AT-specific fluorochrome (DAPI) in the W sex chromosome [12,30]. All the karyotype markers of the two cryptic forms considered allow us to make the suggestion that the submetacentric W sex chromosome resulted from a pericentric inversion of the acrocentric W sex chromosome [12]. Thus, we can see that, in the evolution of *Z. vivipara*, the formation of the viviparous cryptic western form (Z_1_Z_2_W, W-SV) has been accompanied by the changing of the W sex chromosome. It should also be noted that the mechanisms and steps of chromosomal changes in W sex chromosomes for all the described cryptic forms and subspecies of the *Z. vivipara* complex include heterochromatinization event, deletion, tandem fusion, and inversion. These and other mechanisms have also been described in other lizard groups [1,3].

As indicated earlier, alterations in sex chromosomes (and especially in multiple sex chromosomes) are important for evolution, and they may influence the process of meiosis and play a role in isolation and a female meiotic drive (unequal transmission of Z and W chromosomes). Moreover, nonrandom segregation of chromosomes of different morphology (acrocentric versus metacentric) during female meiosis has been documented in birds and mammals [31,32].

Therefore, we reviewed some characteristics of the spermatogenesis and of the oogenesis and early meiosis of two cryptic chromosomal forms of the *Z. vivipara* complex (the eastern (Russian) form and the western form) that have a similar system (Z_1_Z_1_Z_2_Z_2_/Z_1_Z_2_W) but different morphology, and different cytogenetic and genomic structure in the W sex chromosomes [25,26,30,33,34].

## 3. Characteristics of Meiosis in Spermatogenesis and Early Oogenesis

The male diploid karyotype of these forms of *Z. vivipara* is 2n = 36 A: 32 A + Z_1_Z_1_Z_2_Z_2_ (pairs 5 or 6 Z_1_ and 13 Z_2_), and the haploid number is equal to 18 (*n* = 18). During their early meiosis, the synaptonemal complex (SC) bivalents at the prophase I meiosis (the late zygotene–middle pachytene stage and the middle pachytene stage) were found. All SCs did not form asymmetric configuration, and they appeared to be successfully synaptic, including SC Z_1_Z_1_Z_2_Z_2_ sex chromosomes according to the lengths of SC bivalents. However, a wave-shaped morphology of the sex bivalent SC (fifth to sixth in length in a karyotype) was noted [25]. At the diakinesis stage of prophase I meiosis, 18 bivalents were also discovered, including the sex bivalents, without any disruptions in chromosome conjugation (Figure 2A,B). All bivalents were represented by cross-shaped, ring-shaped, or baculiform figures. Their regular segregation with the formation of haploid spermatocytes, 18 chromosomes, at the metaphase II stage of meiosis was constantly revealed [25,34]. The obtained results demonstrated the standard course of meiosis, with formation of constant-haploid-number chromosomes (*n* = 18) in spermatocytes. No clear disturbances in the segregation of chromosomes were detected and the results suggest the stability of their male meiosis.

The female diploid karyotype of both cryptic forms of *Z. vivipara* is 35, 2*n* = 35: 32 A + Z_1_Z_2_W, but in the eastern (Russian) form the W sex chromosome has an acro/subtelocentric (A/ST) shape, whereas in the western form the W sex chromosome has a submetacentric (SV) shape (as a result of a pericentric inversion) (Figure 3A,B) [12]. The ovarian lumen germinal vesicles (oocytes), as well as germinal lamina cells of the females (of the eastern (Russian) form) were examined. During oogenesis, primary follicles enter the early stages of the meiotic prophase I and some characteristics of early oocytes during the early stages of prophase I meiosis (from leptotene to diplotene), synaptonemal complexes (SCs), and lampbrush chromosomes were revealed (Figure 3C,D) [30]. The obtained results demonstrated the standard course of early oogenesis and early meiosis (with formation of a constant number of SC configurations.)

It should be stressed that in meiosis of a female with a diploid chromosome number equal to 35, with multiple sex chromosomes and with the indicated cytogenetical chromosome structure (2*n* = 35: 32 A + 3 sex Z_1_Z_2_W chromosomes), at meiotic prophase I, 16 autobivalents and a complex trivalent of sex chromosomes or complex bivalent and univalent or univalents could be expected. On the basis of light microscopic analysis of the oocyte SCs, and taking into account their length in a female, SC analysis showed that oocytes of these females contained 19 fully synaptic SC elements. The results of previous studies [25] supported the correlation between the morphometric characteristics of relative SC lengths in meiotic prophase I and the metaphase chromosome lengths in somatic cells. Because of this, the female 19 SC elements were assembled and numbered according to their length in descending order (Figure 3C) [26]. The sex chromosome W, on the basis of its size and cytogenetical structure (distribution of C-bands, C/DAPI/CMA3 structure), was attributed to the chromosome pair 5–6 [11]. It should be stressed that SCs were visualized at the stages of late zygotene–middle pachytene. Neither asymmetric configuration nor complex units during the female meiotic prophase I were noticed, only successful synaptic SC elements without any asymmetric configuration [26]. In the eastern (Russian) form, the SC number was constantly equal to 19 (16 SC autosomal bivalents plus 3 SC elements). According to the SC lengths, three SC elements might be univalent of three sex chromosomes Z_1_Z_2_W, or one SC bivalent of W and Z_1_ sex chromosomes and univalent of Z_2_ chromosome and B chromosome univalent. As a result, during meiosis in these female viviparous lizards, 19 SC elements might be formed [26].

During oogenesis and meiosis in the female of the cryptic western form (with reorganized morphological and different cytogenetic structure (C/DAPI/CMA3) of the submetacentric W sex chromosome), primary follicles also entered the early stages of the prophase I of meiosis (stages from leptotene to diplotene) and SC bivalents and lampbrush chromosomes were formed. Again, at these early stages of meiosis (the late zygotene–middle pachytene), neither asynaptic SC configurations nor complex configurations were identified, and the exact SC number has been difficult to count thus far. Nevertheless, at the stages of anaphase–telophase I of meiosis, only rare cells (2 out of 20) with some disturbance in the segregation of bivalents have been revealed [34]. Moreover, in the metaphase II oocytes, haploid numbers equal to 17 with the W sex chromosome (SV) (*n* = 17) were previously determined [35].

Thus, in the females of two cryptic forms of the *Z. vivipara* complex that have a similar system of multiple sex chromosomes (Z_1_Z_2_W) but different morphology, differences also in cytogenetical structure (distribution of heterochromatin, of C/DAPI/CMA3 elements) of the W sex chromosome, resulting from reorganization, early oogenesis and meiosis, have standard occurrence. At the same time, the ambiguous behavior of the three presumed sex chromosomes in the eastern (Russian) form and the lack of variability in their number has been indicated, and in both forms there is a lack of significant disturbances in the chromosome segregation. As is known, the studied populations of *Z. vivipara* in nature do not show the disturbances in sex ratio caused probably by unequal transmission in female multiple sex chromosomes. The obtained results suggest that, along with the shape and cytogenetic structure, there should be a factor (or several factors) that maintains the multiple sex chromosomes and their equal transmission.

As mentioned above, two different short interspersed elements (SINEs) and transposable elements (TE), described in the genome of *Z. vivipara*, have been identified (SINE-Zv 700 and SINE-Zv 300) [24]. SINE-Zv 700 appears to be restricted to *Z. vivipara*, whereas SINE-Zv 300 (including the Gypsy-like fragment) appears to be conserved in many different Squamata species. The active role of the SINEs and the Gypsy-like element in the genomic evolution and differentiation of the *Z. vivipara* complex has been suggested [24]. It is well known that the effects of TEs on the origin of new species are widely discussed in the literature. The activity of TEs might lead to genomic changes, and genetic and phenotypic diversity, often due to new specific gene regulations [36]. TEs are considered by some researchers as potential causes of reproductive isolation across a diversity of taxa [37]. This assumption is associated with the suggestion of some researchers [38] that in reptiles the evolution of sex chromosomes seems to be also explained by some molecular mechanisms, such as gene regulatory mechanisms and others.

It should also be highlighted that fluorescence in situ hybridizations showed a preferential localization of SINE-Zv sequences in the peritelomeric regions of almost all chromosomes, except for the W sex chromosome [24]. The centromere and telomere regions, as is known, are often of key importance for the spatial orientation of chromosomes in the nucleus and are very important for the coincidence of the sites of communication in the hybrid or reorganized (rearrangement) chromosomes with the nuclear envelope, as well as for the conjugation and segregation of chromosomes during meiosis [39]. Apart from this, both centromere and telomere regions play an important role in female meiotic drive [40]. It may be assumed that specific cytogenetic and genomic composition of the W sex chromosome, and the SINE-Zv sequences in the peritelomeric region of chromosomes, might play a role in the meiotic process and the behavior of the sex chromosomes of *Z. vivipara.*

## 4. Conclusions

It becomes clear that in the *Z. vivipara* complex, two closely related viviparous cryptic chromosomal forms (the Eastern (Russian) and the Western), with similar karyotype (male 36/35 female) and system of multiple sex chromosomes (Z_1_Z_1_Z_2_Z_2_/Z_1_Z_2_W) but with different morphology and cytogenetic and specific molecular structure (genomic composition) of their W sex chromosome, demonstrate the standard course of early oogenesis and of female and male meiosis.

In male meiosis, no clear disturbances in the segregation of chromosomes were observed, and the results suggest the stability of male meiosis with formation of a constant number of haploid spermatocytes (*n* = 18).

In female meiosis, during the early stages of prophase I meiosis, the synapted SC bivalents and the lampbrush chromosomes were formed. The SC elements appeared to be fully synapted at the pachytene stage, and no asynaptic SC configurations were observed. The number of SC elements was equal to 19 (in the eastern (Russian) form); however, no significant disturbances, including chromosomal segregation at the anaphase–telophase I stage, were revealed, and haploid oocytes with 17 chromosomes (*n* = 17) were found.

The characteristics of early oogenesis and early meiosis in these two forms of *Z. vivipara* show them maintaining the course of their meiosis and the segregation of multiple sex chromosomes. The question of female meiotic drive in the meiosis of the cryptic forms of the *Z. vivipara* complex is still obscure.

Future studies on genome and karyotype, meiosis, behavior and segregation of multiple sex chromosomes, and their molecular composition, in particular of the centromere/telomere regions, may help to clarify the factors behind the plasticity and the preservation of stability, and the maintenance of high genetic diversity and sex ratio, in the cryptic *Z. vivipara* complex with multiple sex chromosomes.

## Figures and Tables

**Figure 1 animals-13-00019-f001:**
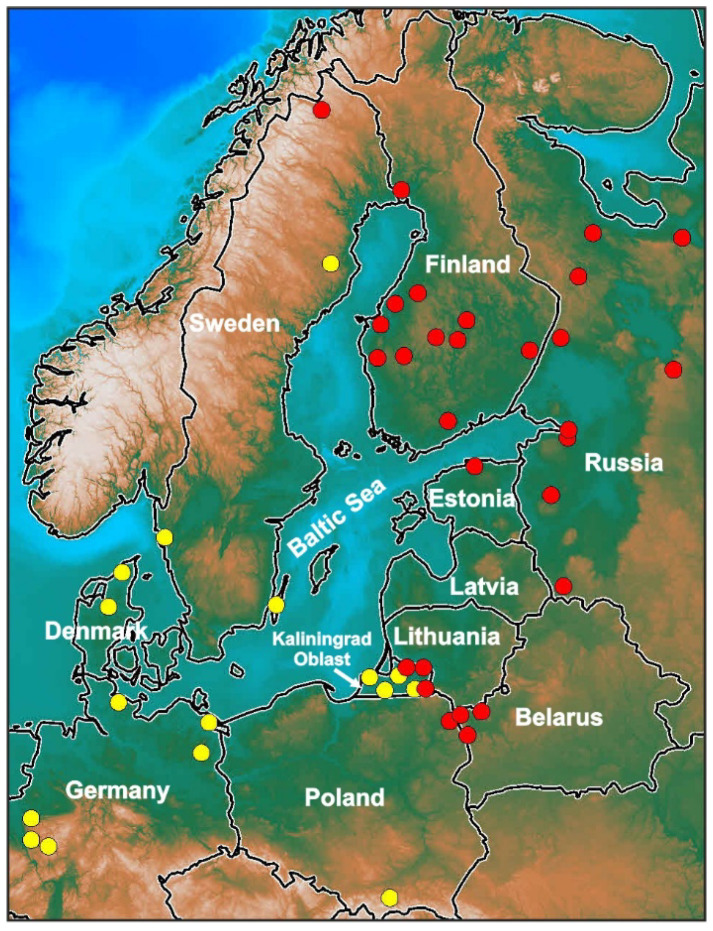
Map showing the distribution of eastern (Russian) (●) and western (●) forms of *Zootoca vivipara* in the Baltic Sea basin based mainly on their karyotypes. Topography is adapted from the GEBCO world map 2014. The points of distribution are from reference [18] with additions.

**Figure 2 animals-13-00019-f002:**
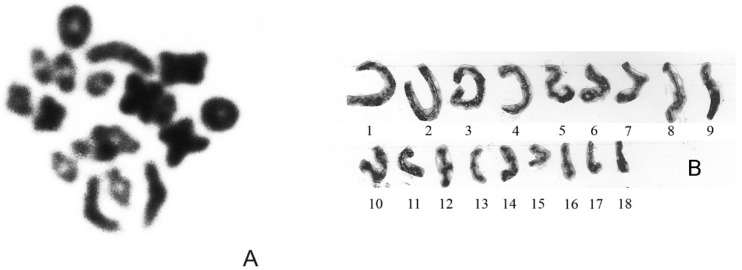
Cells of male specimens of eastern (Russian) cryptic form of viviparous lizard *Zootoca vivipara*: (**A**)—meiotic testis cell at the diakinesis stage; bivalents, *n* = 18; (**B**)—synaptonemal complex (SC) karyotype of spermatocytes, *n* = 18. (**B**) from reference [30].

**Figure 3 animals-13-00019-f003:**
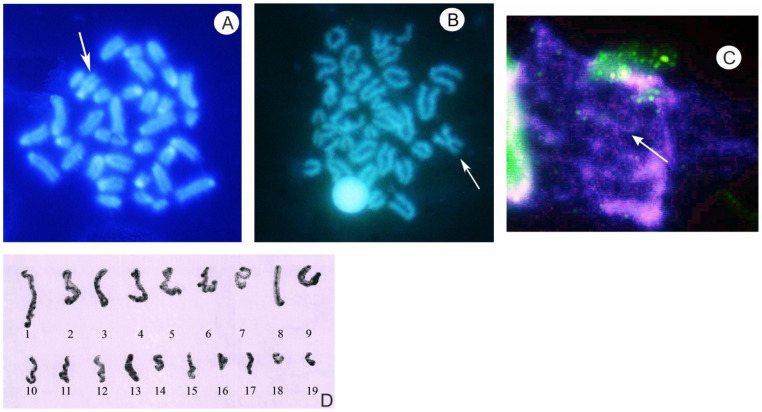
Cells of female specimens of eastern (Russian) form and of western form of viviparous lizard *Zootoca vivipara*: (**A**)—metaphase plate of eastern (Russian) form, specific DAPI stained, 2*n* = 35: 32 A + Z_1_Z_2_W. Arrow points to centromeric and interstitial DAPI blocks of acrocentric W sex chromosome. (**A**) from reference [30]; (**B**)—metaphase plate of western form, specific DAPI stained, 2*n* =35: 32 A + Z_1_Z_2_W. Arrow points to centromeric and weak interstitial DAPI blocks of submetacentric W sex chromosome; (**C**)—the spread oocyte nuclei of female of eastern (Russian) form at pachytene–diplotene stages. Incubation with antibody (SYPC3) and after incubation specific fluorochrome AT DAPI stained. Arrow points to the lampbrush chromosomes. (**C**) from reference [33]; (**D**)—SC karyotype of female eastern (Russian) form, *n* = 16 autosomal bivalents and 3 SC elements of presumed Z_1_Z_2_W sex chromosomes. (**D**) from reference [30].

**Table 1 animals-13-00019-t001:** Karyotype of *Zootoca vivipara*, subspecies, and chromosomal forms with characteristics of sex chromosomes system, morphology of w/W chromosomes, reproductive modality and distribution.

№	2n ♂/♀	System of Sex Chromosomes ♂/♀	Morphology of w/W Sex Chromosomes	Mode of Reproduction, O/V (Ovi-/Viviparous)	Localities	Species, Subspecies, Chromosomal Forms
The first group of karyotype						
1.	36A/36A	ZZ/Zw	M	O	Central, southwestern Europe	*Z. vivipara*, now *Z. carniolica*
2.	36A/36A	ZZ/Zw	M	V	Central Europe	*Z. vivipara*, now *Z. vivipara* Hungarian form
The second group of karyotype						
3.	36A/35 (34A + 1 A/ST)	Z_1_Z_1_Z_2_Z_2_/Z_1_Z_2_W	A, ST	O	Western Europe	*Z. vivipara*, now *Z. v. louislantzi* Pyrenean form
4.	36A/35 (34 A + 1 A/ST)	Z_1_Z_1_Z_2_Z_2_/Z_1_Z_2_W	A/ST	V	Central Europe	*Z. vivipara*, now *Z. vivipara* Austrian form?
5.	36A/35 (34A + 1 SV)	Z_1_Z_1_Z_2_Z_2_/Z_1_Z_2_W	SV/ST	V	Central Europe	*Z. vivipara*, now *Z. vivipara* Romanian form
6.	36A/35 (34A + 1 A/ST)	Z_1_Z_1_Z_2_Z_2_/Z_1_Z_2_W	A/ST	V	Eastern Europe, eastern Baltic region, Asia	*Z. vivipara*, now *Z. vivipara* Eastern (Russian) form
7.	36A/35 (34A + 1 SV)	Z_1_Z_1_Z_2_Z_2_/Z_1_Z_2_W	SV	V	Western, Central Europe, Baltic region	*Z. vivipara*, now *Z. vivipara* Western form

Characteristics of *Z. vivipara:* karyotype and system of sex chromosomes: ZZ/Zw—simple system; ZZ_1_ZZ_2_/Z_1_Z_2_W—multiple sex chromosome system; shape and morphology of w/W sex chromosomes: w—microchromosome, W—macrochromosome; A—acrocentric, ST—subtelocentric, SV—submetacentric; reproductive modality: O—oviparous; V—viviparous; distribution area.

## Data Availability

The data considered in this work can be found in the manuscript, in Table 1.

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
