# Peer review of "A Brief Review of Meiotic Chromosomes in Early Spermatogenesis and Oogenesis and Mitotic Chromosomes in the Viviparous Lizard Zootoca vivipara (Squamata: Lacertidae) with Multiple Sex Chromosomes"

_animals, 2022, doi:10.3390/ani13010019_

Round 1
Reviewer 1 Report
The manuscript titled “A Brief review of meiosis in spermato - and early oogenesis in 2 the viviparous lizard Zootoca vivipara (Squamata: Lacertidae) 3 with multiple sex chromosomes” is in a good presentation and contains very relevant study. Unfortunately,I don’t think this is a relevant paper to publish at Animals. The authors just use other papers as references to the study. Even in a review it is expected that the authors have been produced some investigative material such as comparison charts or some comparative images. Nevertheless, this is an important data for the understanding of herpetofauna chromosomes, specifically, to the Squamata, Lacertidae family. The “brief review” could be published as a note in a new journal with more details. So, I Reject the manuscript but I encourage the authors to submit to another journal as a note.
Author Response
The manuscript titled “A Brief review of meiosis in spermato - and early oogenesis in 2 the viviparous lizard Zootoca vivipara (Squamata: Lacertidae) with multiple sex chromosomes” is in a good presentation and contains very relevant study. Unfortunately, I don’t think this is a relevant paper to publish at Animals. The authors just use other papers as references to the study. Even in a review it is expected that the authors have been produced some investigative material such as comparison charts or some comparative images. Nevertheless, this is an important data for the understanding of herpetofauna chromosomes, specifically, to the Squamata, Lacertidae family. The “brief review” could be published as a note in a new journal with more details. So, I Reject the manuscript but I encourage the authors to submit to another journal as a note. |
We sincerely appreciate your assistance with our manuscript.
The authors included more information about the investigative material and several methods (p 3-4, lines 115-139). “Several samples of Z. vivipara under study had already been used in other studies. Chromosomal material and the preparations from oocytes were stored in a freeze (minus 22-25°C) and some of them were used in this work for the first time. The species is not included in the national Red Data Book and lists of protected taxa. It is not included in international agreements. The Specimens were treated by ether according to ethical practices and were deposited in collection of ZISP, chromosomal collection, accession numbers â„–â„– 9261-9263; 9448-9450”. The authors collected those lizards in the Baltic Sea basin of Russia and added the map of the region in their new text. Some fixed material was used and treated, therefore some new data were included in a discussion as well as two new photos of chromosomes of the species (Figure 2, 3). |
Reviewer 2 Report
In this study the authors reviewed the meiosis in spermato - and early oogenesis in the viviparous lizard Zootoca vivipara. The study falls into the scope of Animals, thus it could be published. Before this can be done, a major revision should be made as specified below.
The manuscript would be beneficed by some figures. For instance, instead to cite “Figures 1,3,5 from Reference [27]”, please, add some figures to this review. The reference 27 is from the same authors, so, I am sure that they can find some interesting figures.
I general, several parts of the manuscript should be better explored.
I am not native English speaker, but I can see that the English must be revised.
Minor issues
Lines 43-44: Lizards are one of a few groups whose members are characterized by temperature (TSD) and genetic (GSD) sex determination. Please, add references.
Lines 44-46: Besides, male and female heterogamety (XY/ZW) and a wide variety of mechanisms of sex determination are also found in these animals. Please, add references.
Line 46: About 25 % of karyotyped lizard species have multiple sex chromosomes.
Please, add references.
Line 50. In contrast, birds (ZW) do not display multiple sex chromosomes [3]. In fact most avian species do not display multiple sex chromosomes, except Pygoscelis adeliae. Reference: Kretschmer, R.; Ferguson-Smith, M.A.; De Oliveira, E.H.C. Karyotype Evolution in Birds: From Conventional Staining to Chromosome Painting. Genes 2018, 9, 181. https://doi.org/10.3390/genes9040181
Lines 99-101 needs reference.
Line 109: change to “tandem fusion, and inversion”.
Line 117: change to “we reviewed”.
Lines 181-184: Please, explore this data, specifically, the “role in the genomic evolution”.
217-219: Redundant: For this reason, today we may assume that the absence of the SINEs and TE elements in THE W SEX CHROMOSOME MAY BE ONE OF THE FACTORS SERVING FOR MEIOTIC STABILITY. THIS W SEX CHROMOSOME MAY BE ONE OF THE FACTORS SERVING FOR MEIOTIC STABILITY.
Please, check the journal reference style and use italic format to scientific names.
Author Response
In this study the authors reviewed the meiosis in spermato - and early oogenesis in the viviparous lizard Zootoca vivipara. The study falls into the scope of Animals, thus it could be published. Before this can be done, a major revision should be made as specified below. The manuscript would be beneficed by some figures. For instance, instead to cite “Figures 1,3,5 from Reference [27]”, please, add some figures to this review. The reference 27 is from the same authors, so, I am sure that they can find some interesting figures. I general, several parts of the manuscript should be better explored. I am not native English speaker, but I can see that the English must be revised.
|
WE are really grateful for your help and support with the manuscript and the possibility to extend the submission deadline. We included our changing here and in a special table
We added some photos of chromosomes during meiotic prophase 1(at the stages of middle pachytene , of diplotene) as well as some metaphase plate after staining with AT specific fluorochrome (DAPI) And we added more information in the text |
Lines 43-44: Lizards are one of a few groups whose members are characterized by temperature (TSD) and genetic (GSD) sex determination. Please, add references. |
References are added. Line 54 |
Lines 44-46: Besides, male and female heterogamety (XY/ZW) and a wide variety of mechanisms of sex determination are also found in these animals. Please, add references. |
References are added. Line 56 |
Line 46: About 25 % of karyotyped lizard species have multiple sex chromosomes. Please, add references. |
References are added. Line 57 |
Line 50. In contrast, birds (ZW) do not display multiple sex chromosomes [3]. In fact most avian species do not display multiple sex chromosomes, except Pygoscelis adeliae. Reference: Kretschmer, R.; Ferguson-Smith, M.A.; De Oliveira, E.H.C. Karyotype Evolution in Birds: From Conventional Staining to Chromosome Painting. Genes 2018, 9, 181. https://doi.org/10.3390/genes9040181 |
Added Line 62 and in the References |
Lines 99-101 needs reference. |
Added Line 162 |
Line 109: change to “tandem fusion, and inversion” |
Done |
Line 117: change to “we reviewed” |
Done Line 168 |
Lines 181-184: Please, explore this data, specifically, the “role in the genomic evolution” |
Added Line 248-252 |
Lines 217-219: Redundant: For this reason, today we may assume that the absence of the SINEs and TE elements in THE W SEX CHROMOSOME MAY BE ONE OF THE FACTORS SERVING FOR MEIOTIC STABILITY. THIS W SEX CHROMOSOME MAY BE ONE OF THE FACTORS SERVING FOR MEIOTIC STABILITY. |
Changed and added Line 264-266 |
Reviewer 3 Report
The authors review the karyological variation of the viviparous lizard Zootoca vivipara, with special emphasis on the meiotic chromosomes of two cryptic forms, the eastern (Russian) form and the western form which possess autosomal and Z chromosomes, but different W chromosome morphologies. They review their previous work of the Russian forms of the species including the meiosis in males and females.
The work is worth to be published but several points should be met before:
First, extensive editing of the English language. Because of the many grammatical mistakes and incorrect expressions this reviewer had to go through several previous publications (including one of the authors) to actually understand the cytogenetics of the species and then review the present work. It was impossible for me to extract any meaningful information from the Simple summary. It is suggested that the authors seek help from a Russian colleague, with knowledge in cytogenetics and good English knowledge to help them to rewrite the work.
I attached the pdf file with some specific comments.
Second, the manuscript would be greatly benefited with some images, for example a map where the cytotypes of Z. vivipara throughout Europe are explained, unless the review is part of a special issue of the Journal and this is covered by other works. W chromosome variations could be clarified by drawings showing the differences in heterochromatin content.
Third, the speculation that short interspersed elements (SINEs) has a role in the regular transmission of the multiple sex chromosomes is based on insufficient evidence and should be eliminated. The reasons for this request are the following:
1) The authors failed to identify the sex chromosomes during meiotic prophase I in females as they explain in page 4 lines 163-164 (The SCs number was difficult to count, at present we can not indicate the exact number of SC bivalents in their oocytes). The fact that they could not identify a trivalent, a bivalent + a univalent or three univalents, does not imply that another mechanism different than regular synapsis is involved in the maintenance of the multiple sex chromosomes in Z. vivipara. That is, negative results cannot be invoked to support a different hypothesis.
2) No arguments are presented on how the absence of a SINE element can regulate the behavior of a chromosome during meiosis, in this case the W chromosomes. Besides, the SINE sequence could have varied on the W chromosome and this could explain lack of hybridization in FISH experiments. That is, the absence of evidence is not evidence for absence.
Finally, The frequency of multiple sex chromosomes derived from ZZ/ZW systems in lizards should be discussed to provide better context about multiple sex chromosmes systems in female, that are comparatively less frequent that XX/XY derived systems.

Author Response
First, extensive editing of the English language. Because of the many grammatical mistakes and incorrect expressions this reviewer had to go through several previous publications (including one of the authors) to actually understand the cytogenetics of the species and then review the present work. It was impossible for me to extract any meaningful information from the Simple summary. It is suggested that the authors seek help from a Russian colleague, with knowledge in cytogenetics and good English knowledge to help them to rewrite the work. |
Authors highly appreciate your hard work. All necessary changes and corrections were made according to your comments and recommendations We sincerely appreciate your assistance with our article and we are very grateful for the opportunity to submit the article. |
Second, the manuscript would be greatly benefited with some images, for example a map where the cytotypes of Z. vivipara throughout Europe are explained, unless the review is part of a special issue of the Journal and this is covered by other works. W chromosome variations could be clarified by drawings showing the differences in heterochromatin content. |
Authors added two Tables, Map and Figures 1 and 2 |
Third, the speculation that short interspersed elements (SINEs) has a role in the regular transmission of the multiple sex chromosomes is based on insufficient evidence and should be eliminated. The reasons for this request are the following |
Authors deleted this part |
The authors failed to identify the sex chromosomes during meiotic prophase I in females as they explain in page 4 lines 163-164 (The SCs number was difficult to count, at present we can not indicate the exact number of SC bivalents in their oocytes). The fact that they could not identify a trivalent, a bivalent + a univalent or three univalents, does not imply that another mechanism different than regular synapsis is involved in the maintenance of the multiple sex chromosomes in Z. vivipara. That is, negative results cannot be invoked to support a different hypothesis. |
Authors agree that further analyses are still needed. Our investigations are still in progress, in particular, the course of oogenesis in different forms. To clarify a situation authors added some data in the text (Lines 134-138 ; 203-2018) to stress a lack of asynaptic SC configurations. “ The lengths of SCs of bivalents were measured using Leica Application Suite V3 on the digital microphotographs. SCs of the bivalents in a karyotype were numbered in the decreasing order of their linear sizes. Analysis of the photos and SC karyotyping were conducted on the basis of the measurements of SC by the relative lengths of each individual SC” |
No arguments are presented on how the absence of a SINE element can regulate the behavior of a chromosome during meiosis, in this case the W chromosomes. Besides, the SINE sequence could have varied on the W chromosome and this could explain lack of hybridization in FISH experiments. That is, the absence of evidence is not evidence for absence. |
Author agree, they deleted this part |
Finally, The frequency of multiple sex chromosomes derived from ZZ/ZW systems in lizards should be discussed to provide better context about multiple sex chromosomes systems in female, that are comparatively less frequent that XX/XY derived systems. |
In a brief review authors have not concerned the special questions of arising of multiple sex chromosomes. In the text they noted a presumable mechanism for the species studied by them, Zootoca vivipara (P. 2) |

Round 2
Reviewer 1 Report
The review is better described and have more information than before. Despite not having included new data, figures or tables with novelty, the review is very well written and presents a good information collection. I believe that the way it was revised this time seems to be in a good format to be published in this journal.
Author Response
We are really grateful for your help and support with the article.
Reviewer 2 Report
I am happy with the author's reply. The manuscript has been improved and can be accepted for publication after minor revision.
Line 79: [reference MT]. I think the reference is missing here.
Author Response
We sincerely appreciate your assistance with our article.
Line 79: [reference MT]. I think the reference is missing here.
We added some new references LINE 74.
Reviewer 3 Report
I can see that changes were introduced in order to improve the manuscript, including the English writing. Still, the work is severely affected by the poor English writing and also by several unclear statements throughout the manuscript.
Here´s a list of examples to support my view:
- The title is misleading. The authors include the words meiosis, oogenesis and spermatogenesis in it. The review show data on early meiosis (prophase I), but the processes of gamete formation in males and females, that is, spermatogenesis and oogenesis, are not discussed.
- In the abstract, the phrase: “however, their early spermato- and oogenesis and meiosis have standard occurrences”. The authors probably mean that early spermato- and oogenesis and meiosis proceed normally or without any disturbance.
- Page 5, line 108. Thus, it is clear that karyological differentiation in the Z. vivipara complex is high”.
What it is clear to this reviewer is that the karyotype in this species complex varies only between 2n=35 and 36, and that most autosomes are acrocentric. High karyological differentiation is observed in wild populations of M. musculus, where 2n can vary between 40 to 21, with ten Robertsonian races occurring only in some regions of the Alps (Piálek J, Hauffe HC, Searle JB: Chromosomal variation in the house mouse. Biol J Linn Soc 84:535-563 (2005). In Z vivipara the chromosome variations are mainly limited to the W chromosome. Although interesting, this does not qualify as high karyological variation.
- Page 10, lines 303 and 314. In the first paragraph, the authors refer to the role of telomeres and centromeres in the chromosome orientation within the meiotic nucleus. In the following paragraph they conclude that because those repetitive sequences (in telomeres and centromeres) may influence the chromosome behavior, then the SINE Zv sequence influence the behavior of the sex chromosomes in Z vivipara. This kind of reasoning is incorrect at several levels, to mention one: telomeric and centromeric sequences are not the same as SINEs.
This type of arguments and mistakes are present throughout the manuscript, weakening its scientific value.
– Finally, I would like to mention that the synaptonemal complexes in Figure 3D are not clear at all. It is impossible from the photograph to confirm or deny if the sex chromosomes synapse or not. Counting the elements is not sufficient because the quality of the synaptonemal complex is not good enough.
Author Response
The authors highly appreciate your help in the improvement of our article, and agree with all your comments.
Comments and Suggestions for Authors
I can see that changes were introduced in order to improve the manuscript, including the English writing. Still, the work is severely affected by the poor English writing and also by several unclear statements throughout the manuscript.
Here´s a list of examples to support my view:
1.The title is misleading. The authors include the words meiosis, oogenesis and spermatogenesis in it. The review show data on early meiosis (prophase I), but the processes of gamete formation in males and females, that is, spermatogenesis and oogenesis, are not discussed.
The authors agreed with your remarks, and changed the title. In this text we discussed mainly the questions of meiotic chromosomes (prophase1 of meiosis) in early spermatogenesis and oogenesis, and mitotic chromosomes. Lines 1 – 3 The title: “A brief review of meiotic chromosomes in early spermatogenesis and oogenesis and mitotic chromosomes in the viviparous lizard Zootoca vivipara (Squamata: Lacertidae) with multiple sex chromosomes”
2. the abstract, the phrase: “however, their early spermato- and oogenesis and meiosis have standard occurrences”. The authors probably mean that early spermato- and oogenesis and meiosis proceed normally or without any disturbance.
The authors agreed with your remarks and changed, line 24-26
3. Page 5, line 108. Thus, it is clear that karyological differentiation in the Z. vivipara complex is high”.
What it is clear to this reviewer is that the karyotype in this species complex varies only between 2n=35 and 36, and that most autosomes are acrocentric. High karyological differentiation is observed in wild populations of M. musculus, where 2n can vary between 40 to 21, with ten Robertsonian races occurring only in some regions of the Alps (Piálek J, Hauffe HC, Searle JB: Chromosomal variation in the house mouse. Biol J Linn Soc 84:535-563 (2005). In Z vivipara the chromosome variations are mainly limited to the W chromosome. Although interesting, this does not qualify as high karyological variation.
The authors added some information about the morphology and the cytogenetical structure (the amount and distribution of heterochromatin, C-bands) of their W sex chromosome - Lines 80-82 and Lines 89-90
4. Page 10, lines 303 and 314. In the first paragraph, the authors refer to the role of telomeres and centromeres in the chromosome orientation within the meiotic nucleus. In the following paragraph they conclude that because those repetitive sequences (in telomeres and centromeres) may influence the chromosome behavior, then the SINE Zv sequence influence the behavior of the sex chromosomes in Z vivipara. This kind of reasoning is incorrect at several levels, to mention one: telomeric and centromeric sequences are not the same as SINEs.
This type of arguments and mistakes are present throughout the manuscript, weakening its scientific value.
The authors agree with the reviewer that centromeric and telomeric regions of chromosomes have complex structure of chromosomes. The authors did not discuss this problem in depth but noted that the specific cytogenetic and genomic composition of the W sex chromosome, and the SINE-Zv sequences in the peritelomeric region of chromosomes, might play a role in the meiotic process and the behavior of the sex chromosomes of Z. vivipara. Lines 266-268
5. Finally, I would like to mention that the synaptonemal complexes in Figure 3D are not clear at all. It is impossible from the photograph to confirm or deny if the sex chromosomes synapse or not. Counting the elements is not sufficient because the quality of the synaptonemal complex is not good enough.
The authors added some changing and information. Lines 177-181 and 213-216 Our investigations are still in progress
In their reply they would also like to add that at the stages of late zygotene-middle pachytene SC were visualized and the synapsis of meiotic chromosomes was analyzed. In male meiosis 18 SC bivalents, including sex bivalents, formed along the whole length, were reviewed. All SCs appeared to be successfully synaptic, but a waveshaped morphology of the sex bivalent SC (fifth to sixth in length in a karyotype) was observed. In female meiosis SCs did not formed asymmetric configuration, as well as no complex units etc, only successfully synaptic SC elements were observed. At the stages of middle pachytene-early diplotene the lampbrush chromosomes were found. On the basis of light microscopic analysis and taking into account the length of SC elements staining by silver nitrate solution, SC karyotype, consisting 19 elements (n=19) , was stability identified. We should also like to stress that tree presumably sex chromosomes never formed complex units, configurations etc. remaining perhaps by 3 univalents.